# Prevalence of Pneumococcal Carriage among Jordanian Infants in the First 6 Months of Age, 2008–2016

**DOI:** 10.3390/vaccines9111283

**Published:** 2021-11-05

**Authors:** Adnan Al-Lahham

**Affiliations:** Department of Biomedical Engineering, School of Applied Medical Sciences, German Jordanian University, Amman 11180, Jordan; adnan.lahham@gju.edu.jo; Tel.: +962-799-706-079

**Keywords:** infants, *Streptococcus pneumoniae*, PCVs, carriage, resistance

## Abstract

Background: *Streptococcus pneumoniae* is an opportunistic human-adapted pathogen driven by nasopharyngeal carriage. Aims: To find the pneumococcal carriage rate, resistance, serotypes, and coverage of pneumococcal conjugate vaccines (PCVs) among infants in the first six months of age in the period from March 2008 to April 2016. Methods: Nasopharyngeal swabs (NP) were taken from healthy infants from the northern part of Jordan. Swabs were processed for cultivation, identification, resistance testing and serotyping according to standard methods. Results: During the surveillance period, 484 infants of this age group were tested, with a total carriage rate of 56.2%. 96.2% of infants one to two months of age got one PCV7 injection and were 58% carriers at the time of the first injection. At age three to four months, 84.9% had received two injections, with a carriage rate of 54.9% at the time of the second injection. At ages five to six months, 12.5% had received one to three injections, with a carriage rate of 43.8%. Predominant serotypes in all age groups were 19F (12.5%), 6A (11.4%), 11A (8.4%), 19A (7.0%), 6B (6.6%), 23F (5.9%), 15B (5.1%), 15A and 23A (4.0% each). Coverage of PCV7, PCV13 and the future PCV20 among all cases were 30.5%, 50.7% and 70.6%, respectively. The highest coverage rate of 78.6% was noticed in the age group at five to six months with the future PCV20. Antibiotic resistance was the highest in the first age group. Conclusions: Pneumococcal carriage starts from the first month of the infant’s life. The highest coverage was noticed for PCV20, which implies the necessity for inoculation with future vaccines.

## 1. Introduction

Nasopharyngeal (NP) carriage of *S. pneumoniae* (i.e., pneumococcus) represents the first step leading to invasive infection. As infants, particularly in low- and middle-income countries, are particularly affected by the severe consequences of invasive pneumococcal infection, characterization of carriage rates is significant from a public health point of view [1,2]. People at risk are low age groups with no or low vaccination strategies [3,4,5]. Studies have shown that pneumococcal colonization in infants starts at four to six months of age [3,6]. Carriage increases during the first years of life and reaches a peak of 50–80% in children of two to three years of age [7,8]. Carriage decreases as age increases, showing a 5–10% carriage rate in ages over 10, but then it eventually increases in the elderly [9,10]. *Streptococcus pneumoniae* contributes to more than one million deaths annually worldwide [11], and was named as the forgotten killer in children by the WHO in 2006 [12]. Approximately 100 serotypes of *S. pneumoniae* have to date been identified, and many of them contribute to high resistance and cause invasive diseases [13,14,15]. This resistance to the antibiotics made the treatment more difficult, which is also particular for young children attending the day care centers. On the other hand, emergence of penicillin- and cephalosporin-resistant strains has created an urgent need for pneumococcal vaccines that are effective in infants [16,17]. To date, PCVs known are: PCV7 with serotypes 4, 6B, 9V, 14, 18C, 19F, and 23F; PCV10 with an additional three serotypes (1, 5, 7F); and PCV13 with an additional three more serotypes (3, ^A and 19A). The future PCV20 included seven additional serotypes (8, 10A, 11A, 12F, 15B/C, 22F and 33F). The pneumococcal conjugate vaccine (PCV7) was introduced in the USA in 2000 and then used in the National Immunization Programs (NIPs) of many countries [18]. These PCVs proved to be safe, immunogenic, and efficient in reducing nasopharyngeal carriage. Therefore, it is important to understand the epidemiology of carriage to implement the appropriate vaccination strategy, especially in the developing and low-middle income countries [19]. Jordan is classified as a middle-income country with a total population of 10,288,521, including 196,900 infants in the first year of life as estimated by the statistical department of the Ministry of Health in 2021. Furthermore, the number of births in Jordan per year is estimated 210,240, with a mortality rate of 13.4 deaths/1000 live births. In Jordan, no IPD data has been available to-date. The statistical department of the MOH of Jordan reports the non-meningococcal infections annually but doesn’t disclose the pneumococcal part. The incidence of non-meningococcal infections in Jordan in 2008 was 9.2/100,000, and was decreased to 4.5/100,000 in 2016. Although PCVs were introduced to the Jordanian private sector in 2000, to-date they still have not been included in the NIP of the country. The information about pneumococcal strains found in Jordanian children and in the NP-carriage of infants in the first six months of life is limited. Furthermore, this age group is the most affected group in pneumococcal diseases, which cause invasive and non-invasive infections globally. Thus, this study was undertaken on infants up to the first six months of age in order to find out the serotypes rotating, resistance and the coverage of the pneumococcal conjugate vaccines, and taking into consideration that the future PCV20 might have an excellent impact on Jordanian infants in the future.

## 2. Materials and Methods

Ethical clearance statement: The research project was approved by the Independent Ethical Committee (IEC) from the Ministry of Health (MOH) in Jordan followed by the approval of the MOH (approval number 8/75/2/2257), and approvals of the directorates of each day care center (DCC) taking part in this research project. Informed written consent for the infant’s participation and the use of NP swabs was obtained from parents prior to the study. The parents were educated on the benefits of future vaccination with the available PCVs.

Study Population and Design: Nasopharyngeal swabs were taken from infants up to six months of age from governmental day care centers (DCCs) from the governorates of Ajlun (*n* = 410), Madaba (*n* = 21), and a private clinic in Amman (*n* = 53) with 484 total samples. Only one NP-swab was obtained from each infant. All NP-samples were collected from trained medical doctors of each DCC or private clinic. Positive results of carriage with resistance analysis and serotyping was sent to the medical doctors of each DCC or to the private clinics for registration on files of the participating infants. All these cases visit the DCC and the private clinic for periodic checkups and vaccinations as recommended by the Ministry of Health of Jordan. The period of collecting the NP-swabs and data covered the period from 2008–2016.

Laboratory procedures: NP-swabs were obtained from infants who had no previous antibiotic consumption before obtaining the NP-swabs. Processing of NP-swabs was undertaken at the microbiological labs of the German Jordanian University. The swabs were cultivated on Columbia Agar plates supplemented with 5% sheep blood, as previously described [7,20]. The plates were incubated at 35 °C with 5% CO_2_ overnight. Identification was done by conventional microbiological methods like colony morphology, susceptibility to optochin (bioMérieux) and bile solubility. Confirmed *S. pneumoniae* isolates were tested for the Minimal Inhibitory Concentration (MIC) using the micro broth dilution as recommended by the Clinical Laboratory Standards Institute (CLSI), with breakpoints and interpretation of susceptibility according to the latest CLSI standards [21]. *S. pneumoniae* ATCC 49,619 was used as a control strain. The VITEK2 compact system with Vitek cards (AST03) and E-test provided by bioMérieux were used to test the following antibiotics: penicillin G (PEN), amoxicillin (AMOX), cefotaxime (CETA), cefuroxime (CEFU), cefpodoxim (CEPO), clarithromycin (CLA), clindamycin (CLI), tetracycline (TET), levofloxacin (LEV), moxifloxacin (MOX), telithromycin (TEL), trimethoprim/sulfamethoxazole (SXT), chloramphenicol (CHA), and vancomycin (VAN). Resistant isolates were double-checked by the use of the E-test provided by bioMérieux. Serotyping of the pneumococcal isolates was performed by the Neufeld’s Quellung reaction method using type and factor sera provided by the Statens Serum Institute (SSI), Copenhagen, Denmark.

Statistical analysis: A Student’s t-test was considered for significant differences using two-tailed values with the level of significance at *p* < 0.05. Other analysis include the rate of carriage, vaccine and non-vaccine serotype coverage, and resistance rates to antibiotics.

## 3. Results

Four hundred and eighty four (484) infants aged one to six months of age were enrolled in this study. Table 1 grouped the infants according to month of age, since vaccination schedule and strategy is at two, four and 10 months of age, as recommended by the vaccination committee of the MOH of Jordan. The total carriage rate detected was 56.2%. The highest carriage rate was in the age group at two months with 58.1%, and then decreased to 56.7% in the age of three months and was minimum at six months of age with a carriage rate 37.5%. The same table shows the number of PCV7 doses given to infants at different periods. Here, 97.1 of infants at two months of age have received the first injection of PCV7, showing a VT-carriage rate of 15%, whereas VT-carriage for the non-vaccinated infants of the same age was 66.7%.

Table 2 shows the distribution of all serotypes detected among all infant age groups tested. Predominant serotypes in all age groups were 19F (12.5%), 6A (11.4%), 11A (8.4%), 19A (7.0%), 6B (6.6%), 23F (5.9%), 15B (5.1%), 15A and 23A (4.0% each). Predominant serotypes in the first age group (1–2 months) were 6A (13%), 19F (12.4%) and 19A (9.2%), while predominant serotypes of the second age group (three to four months) were 19F (13.7%), 11A (10.9%), and 6A, 23F, and 16F (6.8% each). In the last age group (five to six months), predominant serotypes were 19F, 6A, and 6B (14.3% each).

In Table 3, coverage rates of PCV7, PCV13 and the future PCV20 for the detected serotypes were 28.6%, 52.4%, and 73% for the first age group, respectively. The second group (three to four months) showed coverage of 31.5% (PCV7), 43.8% (PCV13), and 63% (PCV20). The third group (five to six months) showed coverage of 50% (PCV7), 64.3% (PCV13), and 78.6% (PCV20). Total coverage in all age groups showed 30.5% (PCV7), 50.7% (PCV13), and 70.6% (PCV20). Significantly higher coverage was seen for PCV13 (*p* < 0.05) than for PCV7 and PCV20 than PCV13 (*p* < 0.05).

Figure 1 shows the antibiotic resistance profile for 10 antibiotics for isolates from all age groups. MIC criteria for penicillin resistance are the same as those for non-meningitis, i.e., ≥0.12μg/mL. No resistance was detected for moxifloxacin, levofloxacin, telithromycin, or vancomycin. Resistance to penicillin, cefuroxime, cefpodoxim, clarithromycin, and trimethoprim-sulfamethoxazole was highest for all age groups.

## 4. Discussion

The nasopharynx is the usual source of pneumococci for studying the carriage rate [22]. This study describes nasopharyngeal carriage of *Streptococcus pneumoniae* in infants up to six months of age from Jordan. According to this study, the *S. pneumoniae* carriage rate was found to be 56.2%. According to the division of statistics of the Ministry of Health of Jordan, the total population count estimated in 2021 was 10,262,521, with an average family size of 6.1 members. This average of family members may attribute to the high carriage rate, in addition to other factors like low income, smoking among family members, history of sicknesses, immune deficiency, viral infections and history of antibiotic consumption [3,7,23,24]. In this study, the average number of siblings per household found was 3.7. Previous studies reported the attendance at day care as a main factor causing the increase of the carriage rate of *S. pneumoniae* [25]. However, higher pneumococcal carriage frequencies were observed among participants aged <2 years, and in individuals belonging to indigenous communities lacking established pneumococcal-conjugated vaccine immunization schemes, as found in Southwestern Columbia [26]. The distribution and prevalence of the pneumococcal serotypes and carriage varies geographically, therefore it is essential to perform pneumococcal prevalence studies in different geographical areas [27]. The literature on pneumococcal carriage of infants up to six months of age is limited. A study performed on infants in a rural African area of Gambia in the period from September 2008 to April 2009 found that 31% of infants were colonized with pneumococci by the end of the first week after birth [28]. In the Gambian study, the carriage rate quickly exceeded 95% after two months, and co-colonization with multiple serotypes was consistently observed in over 40% of the infants at each sampling point during the first year of life, with a with mean acquisition time and carriage duration of 38 and 24 days [28]. This Gambian study was done on infants that had received at least one PCV7 injection during the first year of life as compared to this study. The vaccination schedule for PCV7 in Jordan is at two, four, and 10 months, which means that a maximum of two PCV7 doses were given to 88.2% of the infants studied, as presented in Table 2. Early acquisition, with rates exceeding 20% after one week after birth, was due to serotypes 6A, 34, and NTs reflecting a rapid loss of maternally derived immunity [29,30]. Other studies performed among rural Gambian mother-infant pairs found carriage rates of 1.5% at birth, increasing to 77% after two months of age [31]. Compared to this study’s findings, only one NP-sample was taken from each infant, and no multiple serotypes were indicated. Increasing the number of swabs taken from the same infants might have increased the carriage rates, and might have obtained multiple serotypes from the same infant. Further studies performed on 999 infants at five to eight weeks of age found a carriage rate of 15.7%, with 36.9% coverage with the PCV13 [24]. Caesarean birth was associated with high abundance of the *Streptococcus* species in early ages of infants in the nasopharyngeal microbiome as found among Dutch infants [32]. Another study was done on 82 infants followed from birth to two years of age from 1974 to 1975, where an average of 12 visits were done per child, and 573 pneumococcal isolates were found in 79 of the cases [33].

A recent study was done on 450 infants from India and 459 infants from Bangladesh, where nasopharyngeal swabs were collected at a baseline of 18 and 36 weeks after birth. This study found a carriage rate of 48% in the Indian and 54.8% in the Bangladeshi cohort at 18 weeks, which increased to 53% and 64.8%, respectively, at 36 weeks. Vaccine serotypes for the same study was higher in the Indian cohort (17.8% vs. 9.8% for PCV-10 and 26.1% vs. 17.6% for PCV-13), with 6A, 6B, 19F, 23F, and 19A as the common serotypes [34].

Before the introduction of PCV10 in Pakistan, a pneumococcal nasopharyngeal carriage study was performed in two districts and found carriage rates between 73.6–79.5% in infants 3–11 months of age, with 54.3% vaccine types covered [35]. In Jordan, one study was performed on children of the kindergarten and found 11.1% pneumococcal carriage in 45 children in the age group of one to six months [36]. Serotypes that dominated in our study were 19F, 6A, 11A, 6B, 23F, and 15A/B in all infants tested. PCV is known as being highly efficient in preventing serious diseases caused by vaccine serotypes, and it prevents symptomless colonization of the nasopharynx, but in the USA, even after the use of PCV vaccination, more cases were prevented through the indirect effects rather than by vaccine-induced immunity in those vaccinated [37]. In most countries applying vaccination strategies with the PCV7, more than 60% of the serotypes are covered [38,39,40]. The overage rate in our study for all infants up to six months of age was 30.5% when considering the PCV7, and increased to 70.6% for the future PCV20. The fact that PCV7 was introduced in the year 2000 started with a high coverage rate of IPDs worldwide, but coverage has decreased after 10 years of use, which influenced the increase of the serotypes in the vaccination formulations so that production of PCV13 was 2010. Currently, after 11 years of PCV13, PCV15 and PCV20 are in process for production to maintain the high coverage rate of this vaccine.

Continuous surveillance of the susceptibility patterns of *S. pneumoniae* in carriers becomes increasingly important because of the increasing emergence of antibiotic-resistant strains worldwide [41]. The antibiotic susceptibility of *S. pneumoniae* isolated from infants ‘nasopharynx reflected alarming rates of resistance to most of the antibiotics tested. The high rates of resistance to different classes of antibiotics in *S. pneumoniae* in this study are presumably a consequence of antimicrobial consumption and its misuse within the Jordanian population [42]. The role of vaccination strategies worldwide and the increase of serotypes in future vaccination is to overcome the IPDs, decrease carriage rates, decrease the resistance to vaccine types and carriage, and to decrease death rates related to pneumococci.

To our knowledge, this is the first information about the serotypes rotating among infants up to six months of age in Jordan, which shows the need of the future PCV vaccine in the country NIP covering the majority of the carriers acquiring carriage in low age, and in the time vaccination with PCVs is necessary. Local information on capsular types of *S. pneumoniae* causing disease in young children is highly important to guide production of effective conjugate vaccine, but data of invasive pneumococcal strains from Jordan are very limited or not available, which forces us to work on the carriers as an indicative method for the rotating serotypes in the population.

The imitations of this study are the accessing of private data of infants to collect data, which might help in ascertaining the reasons behind the high rates of carriage and resistance in a scientific and reasonable way. Another difficulty was the low number of infants in the third age group, because this age group does not come to the DCC very often. Larger sample size collections are needed in order to obtain robust estimates for more serotypes. Delays in the transportation and delivery of samples collected by the lab for processing also affect the viability of the isolates and prevent the achieving of higher rates of carriage.

## 5. Conclusions

In spite of the geographical area of samples taken from Jordan, infants enrolled in this study showed high carriage rates and resistance. Coverage of the pneumococcal conjugate vaccines was highest with the future PCV20 in all groups. Global perspectives of the future PCVs seems to be effective in all issues against IPDs, resistance, and carriage. As a present research topic in Jordan, this research has pinpointed the importance and benefits of vaccination, and that continuous prevalence studies within local communities for low ages is important to highlight the need and necessity for vaccination and its introduction to the NIP of the country.

## Figures and Tables

**Figure 1 vaccines-09-01283-f001:**
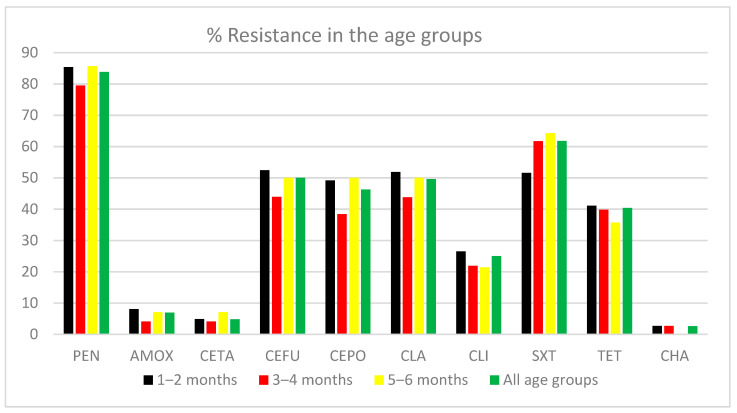
Antibiotic resistance rate in each age group. PEN (Penicillin); AMOX (Amoxicillin); CETA (Cefotaxime); CEFU (Cefuroxime); CEPO (Cefpodoxim); CLA (Clarithromycin); CLI (Clindamycin); SXT (Sulfamethoxazole-Trimethoprim); TET (Tetracycline); CHA (Chloramphenicol).

**Table 1 vaccines-09-01283-t001:** Rate of pneumococcal carriage with VT-carriage for vaccinated and non-vaccinated infants, including number of PCV7 injections in each age group.

Age (M)	No. Cases	Carriers *n* (%)	Cases Received PCV7; No. of Doses *n* (%)	VT-Carriage for Vaccinated Cases *n* (%)	VT-Carriage for Non-Vaccinated Cases *n* (%)
1 M ^a^	4	2 (50.0%)	1 (25%); 1 dose	1/1 (100%)	0/3 (0.0%)
2 M ^b^	315	183 (58.1%)	306 (97.1%); 1 dose	46/306 (15.0%)	6/9 (66.7%)
3 M ^c^	104	59 (56.7%)	93 (89.4%); 2 doses	17/93 (18.3%)	2/11 (18.2%)
4 M ^d^	29	14 (48.3%)	10 (34.5%); 2 doses 2 (6.9%); 1 dose	2/10 (20%) 0/2 (0.0%)	2/17 (11.7%)
5 M ^e^	16	8 (50.0%)	2 (12.5%); 1 dose	0/2 (0.0%)	3/14 (21.4%)
6 M ^f^	16	6 (37.5%)	1 (6.3%); 3 doses 1 (6.3%); 2 doses	0/1 (0.0%) 0/1 (0.0%)	4/14 (28.6%)

^a^ Cases in this age group received a maximum of one dose of PCV7; ^b^ cases received a maximum of one PCV7 dose; ^c^ cases received maximum of 2 PCV7 doses and did not reach four months of age; ^d^ cases received maximum of two PCV7 doses; ^e^ cases received maximum of two PCV7 doses; ^f^ cases received maximum of two PCV7 doses. Vaccination strategy is at two, four and 10 M of age in Jordan. For vaccinated cases, NP-swabs taken at the time of vaccination. VT (PCV7 vaccine type).

**Table 2 vaccines-09-01283-t002:** Number of serotypes detected in each age group, and rate of vaccinated infants for each serotype.

Serotype	All Age Groups *n* (%)	Age 1–2 M *n* (%)	Age 3–4 M *n* (%)	Age 5–6 M *n* (%)	% Vaccinated *
19F	34 (12.5%)	22 (12.4%)	10 (13.7%)	2 (14.3%)	85.3%
6A	31 (11.4%)	24 (13.0%)	5 (6.8%)	2 (14.3%)	96.8%
11A	23 (8.4%)	14 (7.5%)	8 (10.9%)	1 (7.1%)	91.3%
19A	19 (7.0%)	17 (9.2%)	2 (2.7%)	0 (0.0%)	100%
6B	18 (6.6%)	13 (7.0%)	3 (4.1%)	2 (14.3%)	77.8%
23F	16 (5.9%)	10 (5.4%)	5 (6.8%)	1 (7.1%)	68.85
15B	14 (5.1%)	12 (6.5%)	2 (2.7%)	0 (0.0%)	92.8%
15A	11 (4.0%)	7 (3.8%)	3 (4.1%)	1 (7.1%)	90.9%
23A	11 (4.0%)	7 (3.8%)	4 (5.5%)	0 (0.0%)	81.8%
35B	10 (3.7%)	9 (4.8%)	1 (1.4%)	0 (0.0%)	80.0%
14	10 (3.7%)	5 (2.7%)	4 (5.5%)	1 (7.1%)	80.0%
NT	8 (2.9%)	5 (2.7%)	3 (4.1%)	0 (0.0%)	87.5%
33F	7 (2.6%)	4 (2.2%)	2 (2.7%)	1 (7.1%)	85.7%
16F	7 (2.6%)	2 (1.1%)	5 (6.8%)	0 (0.0%)	100%
15C	6 (2.2%)	5 (2.7%)	1 (1.4%)	0 (0.0%)	100%
24F	6 (2.2%)	4 (2.2%)	2 (2.7%)	0 (0.0%)	100%
17F	5 (1.8%)	3 (1.6%)	1 (1.4%)	1 (7.1%)	80.0%
9V	4 (1.5%)	3 (1.6%)	0 (0.0%)	1 (7.1%)	75.0%
3	4 (1.5%)	2 (1.1%)	2 (2.7%)	0 (0.0%)	100%
10A	4 (1.5%)	3 (1.6%)	1 (1.4%)	0 (0.0%)	100%
9N	3 (1.1%)	2 (1.1%)	1 (1.4%)	0 (0.0%)	100%
33A	3 (1.1%)	1 (0.54%)	2 (2.7%)	0 (0.0%)	66.7%
34	3 (1.1%)	1 (0.54%)	1 (1.4%)	1 (7.1%)	66.7%
7B	3 (1.1%)	2 (1.1%)	1 (1.4%)	0 (0.0%)	66.7%
35F	2 (0.7%)	1 (0.54%)	1 (1.4%)	0 (0.0%)	100%
4	1 (0.36%)	0 (0.0%)	1 (1.4%)	0 (0.0%)	100%
13	1 (0.36%)	1 (0.54%)	0 (0.0%)	0 (0.0%)	100%
21	1 (0.36%)	0 (0.0%)	1 (1.4%)	0 (0.0%)	100%
42	1 (0.36%)	1 (0.54%)	0 (0.0%)	0 (0.0%)	100%
10F	1 (0.36%)	1 (0.54%)	0 (0.0%)	0 (0.0%)	100%
16B	1 (0.36%)	0 (0.0%)	1 (1.4%)	0 (0.0%)	100%
28A	1 (0.36%)	1 (0.54%)	0 (0.0%)	0 (0.0%)	100%
35C	1 (0.36%)	1 (0.54%)	0 (0.0%)	0 (0.0%)	100%
7C	1 (0.36%)	1 (0.54%)	0 (0.0%)	0 (0.0%)	100%
7F	1 (0.36%)	1 (0.54%)	0 (0.0%)	0 (0.0%)	100%
Total	272 (100%)	185 (100%)	73 (100%)	14 (100%)	88.2%

* cases of infants carried these serotypes got maximum of 2 PCV7 doses.

**Table 3 vaccines-09-01283-t003:** Coverage of the pneumococcal conjugate vaccines in each age group.

Age Group	PCV7 *n* (%)	PCV13 *n* (%)	PCV20 *n* (%)
1–2 Months	53/185 (28.6%)	97/185 (52.4%)	135/185 (73.0%)
3–4 Months	23/73 (31.5%)	32/73 (43.8%)	46/73 (63.0%)
5–6 Months	7/14 (50.0%)	9/14 (64.3%)	11/14 (78.6%)
All age groups	83/272 (30.5%)	138/272 (50.7%)	192/272 (70.6%)

## Data Availability

Not applicable.

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
