# Peer review of "Prevalence of Pneumococcal Carriage among Jordanian Infants in the First 6 Months of Age, 2008–2016"

_vaccines, 2021, doi:10.3390/vaccines9111283_

Round 1

Reviewer 1 Report

Estimated Dr. Al-Lahhman, Author of the Study "Prevalence of Pneumococcal carriage among Jordanian Infants in the First 6 months of age, 2008 - 2016",

I've read your contribution with great interest, as it could hopefully shed some lights on the hot-topic represented by the carriage of Pneumococcus among youner age groups from a middle-low income country, still characterized by a highly developed healthcare system such as Jordania.

In fact, Authors were able to characterize the representation of main serotypes among the participants, eventually identifying the predominant serotypes (i.e. 19F, 6A, 11A, etc.) suggesting the implementation of PCV13 and PCV20 have the potentiality to starkly impair the spreading and the carriage of the aforementiones serotypes, to date highly circulating.

Unfortunately, the present paper cannot be accepted by Vaccines for several reasons, that affect the study from its roots, and therefore would require (in order to overcome the criticisms of the presente Reviewer, please excuse me for that) a very extensive overhaul of this article before a potential resubmission.

Firstly and most ostentab, the overall quality of the English text is inappropriate, and often impairs the appropriate understanding of the main text. Some examples from the first rows of the paper:

NP carriage by the pneumococcus is a prerequisite step to serious infections worldwide, especially in infants and in low and middle-income countries...

maybe: NP carriage of S pneumoniae (i.e. pnenumococcus) represents the first step leading to invasive infection. As infants, particularly in low- and middle-income countries [? I guess the underlying meaning of following text], are particularly affected by the severe consequences of invasive pneumococcal infection, characterization of carriage rates is particularly significant from a Public Health point of view.

etc.

Moreover, please be aware that across the text as a whole the Authors did use correctly the English verbs, for example: (row 35 following) "... approximately 100 serotypes of S pneumoniae identified ..." --> approximately 100 serotypes of S pneumoniae HAVE BEEN TO DATE identified..." etc.

Second, I have to report the following methodological issues:

a) Authors report on 484 children, but we are totally blind regarding the parent population. How many children were sampled? How many children were assisted by the medical services reporting on their status during the assessed timeframe? In other words, was this one a convenience sample or a preventive power analysis was performed by the Authors? MOreover, how were the children selected? Were all consecutive patients during the assessed timeframe? or else?

b) statistical analysis is not properly reported: Student t-test compares two continuous variables, but the text focuses on categorical ones as you're looking for differences in the distribution of several serotypes. Moreover, the data reporting is inaccurate. More precisely: as PCV impairs or should implair the mucosal colonization by the serotypes covered by the formulation (therefore there is a worlwide effort to increase the number of serotypes included in the various formulates) the key data reporting should address such issue: there were differences among mucosal colonization and carriage of pneumococcus among vaccinated and not-vaccinated individuals? see the attached example for a table 1 replacement. Moreover, also the main effector (i.e. injection taken) may be misleading in the present way you're reporting on. I mean: individuals younger than 2 months of age cannot recived 2 or 3 doses, while individuals of 3-4 months may have received only 1 out of 2 doses, and the very same for individuals 5-6 months of age (none, 1 or 2 instead of 3). Therefore I would focus on the effector of appropriate vaccination schedule, explaining in the tables note that the vaccination schedule is 1 shot before 2nd month, 2 shots before 4th month, 3 shot before 6th month, etc. 

Similarly, Table 2 makes sense only if the data reporting is furtherly dichtomized between vaccinated and not-properly vaccinated individuals.

Eventually, the discussion on the vaccination serotype is quite roughly addressed. The key problem is: as PCV vaccine impairs the serotypes it includes but expands the "space" for uncovered one by a very-darwinian way, does make it sense shift from PCV7 to 13 or even 20 according to the data from Jordania? It seems so, but it must more extensively discussed.

Finally, please discuss more extensively that the data you're reporting on are quite old, more than 10 years for the first specimens that were collected and discussed. Eventually, please be aware that - being the characteristics of the community quite significant for explaining the colonization of NP tract in younger children, an appropriate reporting of data would include also charactestics of the household (e.g. how many siblings, migration background or not, etc). As it lacks from the present report, I think that Authors do not have access to such information, but it should be at least discussed among the most severe shortcomings of the paper.

Author Response

Dear Reviewer,

Thank you so much for the valuable notes and comments for the manuscript, really much appreciated, and for your interest in this topic, especially for Jordan

The following are the answers and the work done according to your comments

Notes, Comments and Suggestions for Authors (Reviewer 1)

Estimated Dr. Al-Lahham, Author of the Study "Prevalence of Pneumococcal carriage among Jordanian Infants in the First 6 months of age, 2008 - 2016",

I've read your contribution with great interest, as it could hopefully shed some lights on the hot-topic represented by the carriage of Pneumococcus among younger age groups from a middle-low income country, still characterized by a highly developed healthcare system such as Jordania.

In fact, Authors were able to characterize the representation of main serotypes among the participants, eventually identifying the predominant serotypes (i.e. 19F, 6A, 11A, etc.) suggesting the implementation of PCV13 and PCV20 have the potentiality to starkly impair the spreading and the carriage of the aforementioned serotypes, to date highly circulating.

Unfortunately, the present paper cannot be accepted by Vaccines for several reasons, that affect the study from its roots, and therefore would require (in order to overcome the criticisms of the present Reviewer, please excuse me for that (No worries, and much appreciated for the valuable comments, thank you) a very extensive overhaul of this article before a potential resubmission.

Point 1. Firstly and most ostentab, the overall quality of the English text is inappropriate, and often impairs the appropriate understanding of the main text. Some examples from the first rows of the paper:

Answer: Regarding the English language, the whole manuscript will be sent to English editing from the Journal at the time of submission of these corrections and will be checked totally.  

NP carriage by the pneumococcus is a prerequisite step to serious infections worldwide, especially in infants and in low and middle-income countries...

maybe: NP carriage of S pneumoniae (i.e. pnenumococcus) represents the first step leading to invasive infection. As infants, particularly in low- and middle-income countries [? I guess the underlying meaning of following text], are particularly affected by the severe consequences of invasive pneumococcal infection, characterization of carriage rates is particularly significant from a Public Health point of view. etc.

Answer: This statement or paragraph was changed and corrected in the manuscript as requested.  

Moreover, please be aware that across the text as a whole the Authors did use correctly the English verbs, for example: (row 35 following) "... approximately 100 serotypes of S pneumoniae identified ..." --> approximately 100 serotypes of S pneumoniae HAVE BEEN TO DATE identified..." etc.

Answer: The sentence corrected accordingly as requested in the manuscript

Point 2:

Second, I have to report the following methodological issues:

  1. Authors report on 484 children, but we are totally blind regarding the parent population. How many children were sampled? How many children were assisted by the medical services reporting on their status during the assessed timeframe? In other words, was this one a convenience sample or a preventive power analysis was performed by the Authors? Moreover, how were the children selected? Were all consecutive patients during the assessed timeframe? or else?

Answer: During the period from 2008-2019, there was 1866 NP samples tested from North Jordan. Among these children, 484 were infants up to 6 months of age (2008-2016). No infants tested afterwards. Children selected were always those visiting the DCC with ages less than 5 years. Children or infants were not patients, but they come with parents for normal check-up, and requesting vaccination according to the program of the MOH. The medical doctor at the DCC took NP-samples (mostly one sample). Samples transferred to the microbiology lab at the University with the data assessed for the cases. Culture, identification, resistance analysis and serotyping performed in the lab.

Point 3:

  1. Statistical analysis is not properly reported: Student t-test compares two continuous variables, but the text focuses on categorical ones as you're looking for differences in the distribution of several serotypes. Moreover, the data reporting is inaccurate. More precisely: as PCV impairs or should impair the mucosal colonization by the serotypes covered by the formulation, (therefore there is a worldwide effort to increase the number of serotypes included in the various formulates) the key data reporting should address such issue: there were differences among mucosal colonization and carriage of pneumococcus among vaccinated and not-vaccinated individuals? See the attached example for a table 1 replacement. Moreover, also the main effector (i.e. injection taken) may be misleading in the present way you're reporting on. I mean: individuals younger than 2 months of age cannot received 2 or 3 doses, while individuals of 3-4 months may have received only 1 out of 2 doses, and the very same for individuals 5-6 months of age (none, 1 or 2 instead of 3). Therefore, I would focus on the effector of appropriate vaccination schedule, explaining in the tables note that the vaccination schedule is 1 shot before 2nd month, 2 shots before 4th month, 3 shot before 6th month, etc. 

Answer: You are very correct, and thank you so much for this correction. Table 1 completely changed according to your correction and advice. All data also added to the text and discussed accordingly. Regarding the statistical analysis, it is only to compare to numbers whether they are significant or not, and was not done for all variants.

Similarly, Table 2 makes sense only if the data reporting is furtherly dichtomized between vaccinated and not-properly vaccinated individuals.

Answer: again, one more column was added to the table with percentage of vaccinated cases for each serotype (much appreciated)

Eventually, the discussion on the vaccination serotype is quite roughly addressed. The key problem is: as PCV vaccine impairs the serotypes it includes but expands the "space" for uncovered one by a very-Darwinian way, does make it sense shift from PCV7 to 13 or even 20 according to the data from Jordania? It seems so, but it must more extensively discussed.

Finally, please discuss more extensively that the data you're reporting on are quite old, more than 10 years for the first specimens that were collected and discussed. Eventually, please be aware that - being the characteristics of the community quite significant for explaining the colonization of NP tract in younger children, an appropriate reporting of data would include also characteristics of the household (e.g. how many siblings, migration background or not, etc). As it lacks from the present report, I think that Authors do not have access to such information, but it should be at least discussed among the most severe shortcomings of the paper.

Answer: As you have noticed that, some of these data are from 2008 (old), and at that time, only PCV7 was available in the private market. There was no PCV13 or PCV20 at that time. Furthermore, more discussion was added about the PCVs. From the infants involved, no migrants (Syrians) are available, since they came 2011 to Jordan. Average number of children per family in this study is 3.7. This number was included in the discussion.

Reviewer 2 Report

Article describing the percentage of pneumococcal carriage in NP samples, the serotype distribution and the antibiotic resistance in children up to 6 months of age in Jordan, between 2008 and 2016.

Although the article is well understood and I am not a native English speaker, I think it would benefit from a language review done by an English speaker.

There are some missing information the would help in the understanding of the paper. Please, include the answer to these 6 questions in the text.

  1. The study was done between 2008 and 2016. PCV10 and PCV13 were available in 2010. During all the study period only PCV7 was privately used in Jordan?
  2. How many children were sampled each year of the study? How many in each center?
  3. What were children’s reasons to attend the DCC?
  4. Were samples collected and transported as recommended by the WHO on 2003 and updated in 2013? If not, please specify.

K.L. O’Brien, H. Nohynek. Report from a WHO Working Group: standard method for detecting upper respiratory carriage of Streptococcus pneumoniae. World Health Organization Pneumococcal Vaccine Trials Carriage Working Group. Pediatr Infect Dis J 2003; 22: e1-e11.

Satzke C, Turner P, Virolainen-Julkunen A, et al. Standard method for detecting upper respiratory carriage of Streptococcus pneumoniae: Updated recommendations from the World Health Organization Pneumococcal Carriage Working Group. Vaccine 2013; 32: 165-179.

  1. How many children carrying PCV7 serotypes had been vaccinated with at least one PCV7 dose (“vaccine failure in preventing carriage”)?
  2. I think the appropriate test to analyze the data should have been the Chi-square test, as the variables studied were categorical (only one analysis show, line 125, coverage yes/not).

Minor comments

  • Line 33-34. Please, include the years of the mortality estimation.
  • Line 111. Check “thev”.
  • Line 120. Check “is were”.
  • Line 124. “… than both PCV7”. Should it say: “…than for PCV7”
  • Line 136. Check “highest”.
  • Please use “S. pneumoniae” all through the discussion, in italics.
  • Line 153. “… higher pneumococcal carriage frequencies … aged < 2years”. Should it say “aged <2 months”?
  • Lines 174-187. Many data from other studies, but no comparison/discussion with data obtained by the author.
  • Line 202. Has the author any reference to the consumption of antibiotics in Jordan?
  • Line 206. The author states that “…this is the first information about the serotypes rotating among infants up to 6 months of age in Jordan”. However, the author has published another article describing serotypes in carriers aged 1 to 163 months between 2015-2019, which are more recent data (Adnan Al-Lahham. Vaccines 2021, 9: 789).
  • Figure 1. I would suggest to show resistance data in a Table as in Table 3 of the previous article (Adnan Al-Lahham. Vaccines 2021) as it shows more specific information about MICs.

Author Response

Although the article is well understood and I am not a native English speaker, I think it would benefit from a language review done by an English speaker.

Answer: The manuscript will be reviewed by and corrected by the Journal English editing

There are some missing information the would help in the understanding of the paper. Please, include the answer to these 6 questions in the text.

  1. The study was done between 2008 and 2016. PCV10 and PCV13 were available in 2010. During all the study period only PCV7 was privately used in Jordan?

Answer: This is true, but there was some PCV7 donations from Pfizer to Jordan and were used in one location (Ajlun) to study the impact of vaccination on carriage. This study was done in 2008, where some of the infants were included

  1. How many children were sampled each year of the study? How many in each center?

The samples were taken from 4 governorates in Jordan (Ajlun 410 cases from 13 centers from 2008-2010, Amman 13 cases from one center 2015-2016, Wadi Alseer 40 cases from one center 2008, and Madaba 21 cases from one center in 2015)

  1. What were children’s reasons to attend the DCC?

Answer: They come for normal check-ups and to get the required vaccination that is given at each DCC by the MOH included in the NIP

  1. Were samples collected and transported as recommended by the WHO on 2003 and updated in 2013? If not, please specify.

Thank you for the support with the literature of O brian and Satzke. I know these papers and we have 100% work in common. our NP-swabs were taken by the medical Doctors, put directly in a transport media from the company COPAN with charcoal for preservation and sent directly with a special driver to the lab for culture. This was also done at the National Reference Center for Streptococci in Aachen -Germany as i worked for 8 years. This method was applied for Europe too. 

K.L. O’Brien, H. Nohynek. Report from a WHO Working Group: standard method for detecting upper respiratory carriage of Streptococcus pneumoniae. World Health Organization Pneumococcal Vaccine Trials Carriage Working Group. Pediatr Infect Dis J 2003; 22: e1-e11.

Satzke C, Turner P, Virolainen-Julkunen A, et al. Standard method for detecting upper respiratory carriage of Streptococcus pneumoniae: Updated recommendations from the World Health Organization Pneumococcal Carriage Working Group. Vaccine 2013; 32: 165-179.

  1. How many children carrying PCV7 serotypes had been vaccinated with at least one PCV7 dose (“vaccine failure in preventing carriage”)?

Answer: I have repeated the table 1 and added all these new information. We cannot say that infants have vaccine failure after one PCV injection. But majority of infants in this study got the first dose of PCV7 at the time of the NP-swab.  

I think the appropriate test to analyze the data should have been the Chi-square test, as the variables studied were categorical (only one analysis show, line 125, coverage yes/not).

Minor comments

  • Line 33-34. Please, include the years of the mortality estimation.

Answer: done

  • Line 111. Check “thev”.

corrected

  • Line 120. Check “is were”.

corrected

  • Line 124. “… than both PCV7”. Should it say: “…than for PCV7”

corrected

  • Line 136. Check “highest”.

checked

  • Please use “S. pneumoniae” all through the discussion, in italics.

checked for all the manuscript

  • Line 153. “… higher pneumococcal carriage frequencies … aged < 2years”. Should it say “aged <2 months”?

No, this was a study done in Columbia for ages less than 2 years and was cited here

  • Lines 174-187. Many data from other studies, but no comparison/discussion with data obtained by the author.

Checked and other data added

  • Line 202. Has the author any reference to the consumption of antibiotics in Jordan?

Sorry, there are no data at all for the antibiotic consumption in Jordan. As i was looking for these data to stress on the fact of very high resistance rates even for other bacteria like S. aureus 

  • Line 206. The author states that “…this is the first information about the serotypes rotating among infants up to 6 months of age in Jordan”. However, the author has published another article describing serotypes in carriers aged 1 to 163 months between 2015-2019, which are more recent data (Adnan Al-Lahham. Vaccines 2021, 9: 789).

This is true. In this last publication published by vaccines, the number of cases or infants does not exceed 73 cases, and the goal was to differentiate Amman as the capital of Jordan with a rural poor area of Madaba in total. But here we are discussing all cases up to 6 months of age  

  • Figure 1. I would suggest to show resistance data in a Table as in Table 3 of the previous article (Adnan Al-Lahham. Vaccines 2021) as it shows more specific information about MICs.

Well i intended to put a table but i think as a figure can show also estimates of numbers *(hope you do not mind)

Thank you so much

Reviewer 3 Report

I read with great interest the paper. The research question is good and also the setting is relevant. Only minor suggestions

  1. Introduction: intrude better the idea of research and why for you and your team this question has a relevant issue in public/global/infectious diseases key role
  2. Methods and results: are very clear. 
  3. Discussion: well done, only if you discuss better the role of concomitant vaccination and the role of resistance
  4. Limitation section: I appreciate it
  5. Conclusion: please give some global health prospectives that came from your interesting paper

Author Response

Reviewer 3

Comments and Suggestions for Authors

I read with great interest the paper. The research question is good and also the setting is relevant. Only minor suggestions

  1. Introduction: intrude better the idea of research and why for you and your team this question has a relevant issue in public/global/infectious diseases key role

Answer: A paragraph at the end of the introduction was added stating the relevance of this issue globally.

  1. Methods and results: are very clear. 

Answer: No comments, thank you

  1. Discussion: well done, only if you discuss better the role of concomitant vaccination and the role of resistance

Answer: This issue was added in the discussion

  1. Limitation section: I appreciate it

Answer: No comments, thank you

  1. Conclusion: please give some global health prospectives that came from your interesting paper

Answer: done, thank you

Reviewer 4 Report

The author analyzed pneumococcal carriage rate, resistance, serotypes, and coverage of pneumococcal conjugate vaccines among 484 infants between March 2008 to April 2016. Overall, the study design was rigorous, and the methodology was appropriate, still, several points of concern should be addressed by the author.

Major points;

  1. Please explain the components of PCV7, 10, 13 and 20 vaccines for readers to understand this paper.
  2. Lines 21-23: The conclusion doesn’t make sense to me. “Start early ages and can be in early age”? What is the evidence to conclude “necessity for vaccination with the future vaccine”?
  3. Please include IPD rate between 2008 and 2016 in children younger than 6 month old. Without this data you can’t discuss “benefits of vaccination (Line 225)”.

Minor points;

  1. Line 111: Typo “thev”.
  2. Line 120: Delete “is”.
  3. Line 169: Replace “our” by “this”.
  4. Line 205: Replace “our” by “my”.

Author Response

Reviewer 4

Comments and Suggestions for Authors

The author analyzed pneumococcal carriage rate, resistance, serotypes, and coverage of pneumococcal conjugate vaccines among 484 infants between March 2008 to April 2016. Overall, the study design was rigorous, and the methodology was appropriate, still, several points of concern should be addressed by the author.

Major points;

  1. Please explain the components of PCV7, 10, 13 and 20 vaccines for readers to understand this paper.

Answer: They were added in the introduction

  1. Lines 21-23: The conclusion doesn’t make sense to me. “Start early ages and can be in early age”? What is the evidence to conclude “necessity for vaccination with the future vaccine”?

Answer: The sentence was corrected, thank you

  1. Please include IPD rate between 2008 and 2016 in children younger than 6 month old. Without this data you can’t discuss “benefits of vaccination (Line 225)”.

 Answer: There are no data up-to-date about the IPDs in Jordan (Sorry for this). But Jordan report the non-meningococcal infections where the pneumococcus is one of the causative agents. I reported in the introduction some of the incidence statistics as requested for the non-meningococcal meningitis.

Minor points;

  1. Line 111: Typo “thev”.

Answer: corrected

  1. Line 120: Delete “is”.

Answer: deleted, thank you

  1. Line 169: Replace “our” by “this”.

Answer: corrected, thank you

  1. Line 205: Replace “our” by “my”.

Answer: corrected

Round 2

Reviewer 1 Report

Estimated Authors of the paper "Prevalence of Pneumococcal Carriage among Jordanian Infants in the First 6 Months of Age, 2008-2016",

I've deeply appreciated the considerable efforts you paid in order to cope with my previous review. Unfortunately, despite the significant improvements, the present paper still remains inappropriate for an eventual acceptance (but compared to the earlier stage, it could be accepted after extensive revisions) for the following reasons:

  1. First at all, the quality of the English text MUST be revised. Some sentences lack the auxiliary verb (i.e. "be" or "have"), being quite confusing. Some examples: 190-191: "Compared to this study findings, only one NP-sample [WAS?] taken from each infant, and no multiple serotypes indicated"; 197-198: "Another study done on infants followed from birth to two years of age from 1974 to 1975, where an average of 12 visits done per child and found 573 pneumococcal isolates in 79 of the case" etc. Without an extensive and (please, understand my point of view!) professional editing, acceptance of this paper is honestly impossible.
  2. Authors have improved the discussion, there's no doubt about it. Still, the potential bias represented by the uneven sampling of the children remains unsolved.
  3. Authors are quite confusing when dealing with PCV. PCV is a conjugated Pneumococcal Vaccine, where the antigens of Pneumococcus are associated to a carrier that improves the immune response. PCV is quite innovative when compared to the previously available Pneumococcal polysaccharide vaccine (PPV), that contain a mixture, unconjugated, of polysaccharide antigens. PPV unfortunately is unable to elicit a valuable immune response against some strains, therefore PCV was ultimately developed, with a progressive expansion of the number of strains included in the formulate. In their Introduction, Authors should provide such information, that explains why PCV is so important for our preventive strategy compared to PPV, etc (see for comparison row 41 following: "To date, PCVs known are..."

Please understand that because of the very poor quality of the English text I'm forced to recommend the rejection of this paper. On the other hand, I'm recommending an extensive revision with professional editing of the text, and after such improvements I'm confident that the present study may be quickly accepted.

Author Response

Dear Reviewer, 

Thank you so much for your valuable comments. The manuscript was submitted to the Journal professional editing as requested with the addition of your notes as found below. 

Best regards 

Comments and Suggestions for Authors

Estimated Authors of the paper "Prevalence of Pneumococcal Carriage among Jordanian Infants in the First 6 Months of Age, 2008-2016",

I've deeply appreciated the considerable efforts you paid in order to cope with my previous review. Unfortunately, despite the significant improvements, the present paper still remains inappropriate for an eventual acceptance (but compared to the earlier stage, it could be accepted after extensive revisions) for the following reasons:

  1. First at all, the quality of the English text MUST be revised. Some sentences lack the auxiliary verb (i.e. "be" or "have"), being quite confusing. Some examples: 190-191: "Compared to this study findings, only one NP-sample [WAS?] taken from each infant, and no multiple serotypes indicated"; 197-198: "Another study done on infants followed from birth to two years of age from 1974 to 1975, where an average of 12 visits done per child and found 573 pneumococcal isolates in 79 of the case" etc. Without an extensive and (please, understand my point of view!) professional editing, acceptance of this paper is honestly impossible.

Answer: Dear honored reviewer, you are very correct. That’s why I have submitted this manuscript to a professional English editing at the journal cite and got this confirmation as an evidence for your convenience. 

Basel, 21 October 2021

Description

Payment confirmation for English editing invoice: English-35976

MDPI confirms that it has received payment of English editing invoiceenglish-35976 (invoice dated21 October 2021)

Amount Received: USD412.76

Date Received 21 October 2021

MDPI

Financial Accounting

St. Alban-Anlage 66

CH–4052 Basel

Switzerland

The manuscript was edited as attached and your comments were also added to the manuscriopt

  1. Authors have improved the discussion, there's no doubt about it. Still, the potential bias represented by the uneven sampling of the children remains unsolved.

Answer: I totally understand your point of view. All of my research projects have started since 2008 in Jordan after I moved back from Germany working there at the National Reference Center for Streptococci for at least 8 years. I got a professorship at the German Jordanian University as an Applied University in Applied Medical Sciences School, where I started from zero and doing all the work alone without any financial support. Wyeth (now Pfizer) supported one Project in 2008-2009, and Pfizer 2017-2019, funded the second. Both approved collecting one NP-samples. In between, I was collecting samples from different areas. I have noticed that the number of infants up to 6 months of age were enough for a publication for its importance, although the samples were in different periods or uneven. These are my reasons for the uneven sampling. The Jordanian MOH lacks proofs of publications in good-ranked journals to include the PCV vaccine at the NIP of the country. Jordan is missing the statistical numbers for antibiotic consumption, and there are no National Reference Centers for bacterial species (Streptococci, Staphylococci, Candida, etc…). Furthermore, no statistical data ever found on IPDs. May be such publications stressing on the numbers would have an influence. Sorry for the long paragraph, but hope that you understand my point of view.   

  1. Authors are quite confusing when dealing with PCV. PCV is a conjugated Pneumococcal Vaccine, where the antigens of Pneumococcus are associated to a carrier that improves the immune response. PCV is quite innovative when compared to the previously available Pneumococcal polysaccharide vaccine (PPV), that contain a mixture, unconjugated, of polysaccharide antigens. PPV unfortunately is unable to elicit a valuable immune response against some strains, therefore PCV was ultimately developed, with a progressive expansion of the number of strains included in the formulate. In their Introduction, Authors should provide such information that explains why PCV is so important for our preventive strategy compared to PPV, etc (see for comparison row 41 following: "To date, PCVs known are..."

Answer: Thank you for such comment. I know exactly the difference between the two different PCVs and PPVs available in the market, since our work with Prof Dr Ralf Rene Reinert at the National Reference Center for Streptococci was mainly on this topic, and we have published several papers on this topic. I thought it is something known to all. But as requested will add these points in the introduction after I receive the English editing from the journal.

Thank you again for the rich comments and much appreciated.

Please understand that because of the very poor quality of the English text I'm forced to recommend the rejection of this paper. On the other hand, I'm recommending an extensive revision with professional editing of the text, and after such improvements I'm confident that the present study may be quickly accepted.

Round 3

Reviewer 1 Report

Estimated Author,

I've no further requests and I'm therefore advocating the acceptance of this article